# Autologous Stem Cells in Achilles Tendinopathy (ASCAT): protocol for a phase IIA, single-centre, proof-of-concept study

Andrew J Goldberg,[1] Razi Zaidi,[2] Deirdre Brooking,[1] Louise Kim,[3] Michelle Korda,[4] Lorenzo Masci,[5] Ruth Green,[1] Paul O'Donnell,[1] Roger Smith[6]

¹UCL Institute of Orthopaedics and Musculoskeletal Science (IOMS), Royal National Orthopaedic Hospital (RNOH), Stanmore, UK
²Princess Royal University Hospital, Orpington, UK
³Joint Research and Enterprise Office, St George's, University of London, London, UK
⁴Queen Mary University of London, London, UK
⁵Institute of Sport Exercise and Health, London, UK
⁶Royal Veterinary College, Hatfield, UK

**Correspondence to**
Andrew J Goldberg;
Andy.Goldberg@nhs.net

## ABSTRACT

**Introduction** Achilles tendinopathy (AT) is a cause of pain and disability affecting both athletes and sedentary individuals. More than 150 000 people in the UK every year suffer from AT.  While there is much preclinical work on the use of stem cells in tendon pathology, there is a scarcity of clinical data looking at the use of mesenchymal stem cells to treat tendon disease and there does not appear to be any studies of the use of autologous cultured mesenchymal stem cells (MSCs) for AT. Our hypothesis is that autologous culture expanded MSCs implanted into an area of mid-portion AT will lead to improved pain-free mechanical function. The current paper presents the protocol for a phase IIa clinical study.

**Methods and analysis** The presented protocol is for a non-commercial, single-arm, open-label, phase IIa proof-of-concept study. The study will recruit 10 participants and will follow them up for 6 months. Included will be patients aged 18–70 years with chronic mid-portion AT who have failed at least 6 months of non-operative management. Participants will have a bone marrow aspirate collected from the posterior iliac crest under either local or general anaesthetic. MSCs will be isolated and expanded from the bone marrow. Four to 6 weeks after the harvest, participants will undergo implantation of the culture expanded MSCs under local anaesthetic and ultrasound guidance. The primary outcome will be safety as defined by the incidence rate of serious adverse reaction. The secondary outcomes will be efficacy as measured by patient-reported outcome measures and radiological outcome using ultrasound techniques.

**Ethics and dissemination** The protocol has been approved by the National Research Ethics Service Committee (London, Harrow; reference 13/LO/1670). Trial findings will be disseminated through peer-reviewed publications and conference presentations.

**Trial registration number** NCT02064062.

## Strengths and limitations of this study

► A first-in-man safety study.
► First study of its kind injecting culture expanded mesenchymal stem cells into mid-portion Achilles tendinopathy.
► Capturing both clinical scores and radiological outcomes.
► Independent assessment using ultrasound tissue characteristics.
► Proof-of-concept safety study; therefore, further research will be required.

an incidence of 2.35 per 1000 people, equivalent to >150 000 people in the UK every year.[1]

This is a degenerative process that leads to pain and dysfunction in middle-aged individuals and sports participants.[2] Tendinopathic tendons lose their shiny appearance and microscopy reveals discontinuous and disorganised collagen fibres. This is associated with mucoid and lipoid degeneration and an increasing number of cells within the tendon tissue, mostly with a fibroblastic or myofibroblastic appearance. Often, there is an abrupt discontinuity of both vascular and myofibroblastic proliferation immediately adjacent to the area of greatest abnormality.[3]

Achilles tendon degeneration is evident as an increased signal on MRI[4] and as hypoechogenic regions on ultrasound investigation.[5] These areas of abnormal imaging correspond with areas of altered collagen fibre structure and increased interfibrillar ground substance, which has been shown to consist of hydrophilic glycosaminoglycans.[4]

Achilles tendinopathy (AT) adversely affect quality of life by causing chronic pain and disability.[6] The source of pain has been associated with neurovascular ingrowth and autocrine/paracrine effect of cellular products (eg, catecholamines, acetylcholine,

## INTRODUCTION

The Achilles tendon is the largest tendon in the body and it plays an important role in the biomechanics of the lower extremity. It can withstand great forces, especially during sporting exercise. The general population has

glutamate) and a reduction in pain is a key indicator of treatment success.[3]

Numerous management strategies exist for AT, such as rest, physiotherapy with eccentric strengthening exercises,[7] extracorporeal shockwave therapy,[8] dry needling,[8] high-volume injections[8] and platelet-rich plasma (PRP) injections.[9] However, apart from eccentric loading exercises none of these therapies has been shown to be more effective than placebo.[8 8]

The use of PRP is controversial. One randomised controlled trial (RCT) looked at PRP versus stromal vascular fraction (SVF) derived from adipose. SVF has been shown to be source of MSCs. They showed that the use of PRP and SVF was safe and that both PRP and SVF improved clinical scores with statistical significance. The use of SVF seemed to provide quicker improvement but at 180 days there was no difference between the two groups.[10] Another RCT demonstrated that the use of PRP had no statistically significant advantage over a saline placebo group.[9 11]

Although the majority of tendon conditions are treated non-operatively, up to 45% of patients at some stage consider surgery.[3] Tendon disorders are a major burden on the National Health Service (NHS). A total of 51 454 operations on tendon disorders were performed in the NHS in 2015/2016 (HESOnline), which is up from 48 765 operations performed in 2004/2005. Given that the majority of tendon problems are treated non-operatively, this high surgical caseload represents just the tip of the iceberg of the real cost to the health service of tendon disease. There is therefore a need for safe and effective non-surgical treatments.

Laboratory-based studies looking at autologous mesenchymal stem cells (MSCs) injections into damaged tendons have shown improved functional and histological outcomes.[12–15] Smith et al published an equine case-control trial comparing autologous bone marrow-derived MSCs and saline injections. The treated tendons exhibited statistically significant improvements in key parameters with normalisation of biomechanical, morphological and compositional parameters.[16] Another study followed up 141 horses postintralesional MSC injection for digital flexor tendinopathy. Two-year follow-up showed no adverse effects of the treatment with no aberrant tissue on histological examination. The re-injury percentage was significantly less than published for national hunt racehorses treated in other ways.[14]

Human studies using MSCs have shown promise in many conditions including heart disease,[17] musculoskeletal disorders,[18] autoimmune diseases[19] and neurological disorders.[20]

With regard to the use of MSCs in human tendons, there are only a handful of reports.

Ellera Gomes et al[21] performed a cohort study in 14 patients who received bone marrow concentrate (BMC) derived from the iliac crest with rotator cuff repair. Twelve months following the procedure, all patients had better clinical scores and MRI showed that tendons had healed.

Hernigou et al[22] performed a comparative study looking at effect of BMC cells on rotator cuff healing postarthroscopic repair. Patients treated with BMC had a significantly increased healing rate and a significant reduction in rerupture. In contrast, Mazzocca et al[23] used BMCs from the humeral head with a similar model of rotator cuff repair, and showed no statistical difference in clinical outcome at a mean of 10.6 months.

Pascual-Garrido et al[24] looked at eight patients with patella tendinopathy. BMCs from the iliac crest were injected into the patella tendon under ultrasound guidance. At a mean follow-up of 5 years, there was significant improvement in most clinical scores. This improvement was seen up to year 2 after which it plateaued. Ultrasound evaluation was at 6 months, which showed that eight of the nine patients had improvement in the grading of the patella tendinopathy.

Singh et al[25] looked at 30 patients with untreated tennis elbow. BMCs from iliac crest were injected into the lateral epicondyle. There was a statistically significant improvement in the patient-rated tennis elbow evaluation score at 12 weeks.

All of these studies used mononuclear cells from bone marrow concentrate. Usuelli et al published a study looking at PRP and SVF. The study concluded that SVF was safe and as efficacious as PRP at 180 days, although this study did not use culture expanded MSCs.[10] We could only identify one study looking at culture expanded MSCs in AT. Ilić et al reported the use of human placenta-derived allogenic MSCs injected under ultrasound guidance in six patients. They monitored for the primary outcome, which was safety. They concluded it was safe to inject human placenta-derived MSCs into human Achilles tendons, however made no comment on clinical outcome.[26]

There is clearly a lack of reliable evidence for the use of MSCs in tendon disease. Our hypothesis is that autologous culture expanded MSCs implanted into a tendon with AT will lead to improvement in pain-free mechanical function. The current paper presents the protocol for phase IIa trial to explore this.

## METHODS AND ANALYSIS
### Study design
Following dialogue and engagement with the Medicines and Healthcare Regulatory Agency (MHRA), the trial was defined as a non-commercial, single-arm, open-label, phase IIa proof-of-concept study. The study will be conducted at the Royal National Orthopaedic Hospital (RNOH). The study will recruit 10 participants and will follow them up for 6 months (figure 1). The MHRA considered this study to be a first-in-man (phase IIa) study whose primary role was to demonstrate safety and hence all patients were to get the active treatment and no control group was used. The information from this study would then be used to design a subsequent phase III RCT involving a control.

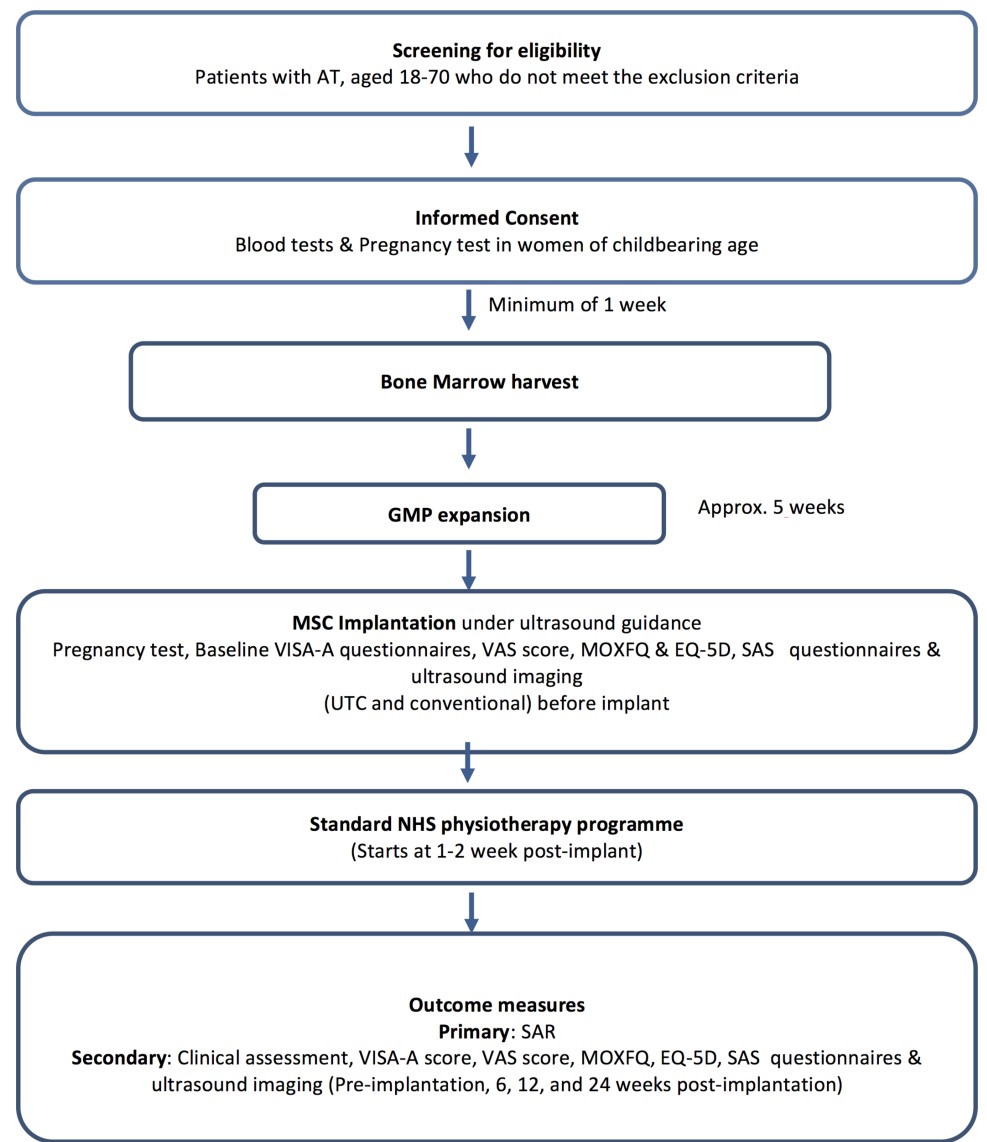

**Figure 1** Trial flow diagram. AT, Achilles tendinopathy; EQ-5D-5L, EuroQol 5-dimension 5-level; GMP, good medical practice; MOXFQ, Manchester Oxford Foot and Ankle Questionnaire; NHS, National Health Service; SAS, Sporting Activity Score; UTC, ultrasound tissue characteristics; VAS, Visual Analogue Score; VISA-A, Victorian Institute of Sports Assessment Achilles Questionnaire.

The primary outcome will be safety as defined by the incidence rate of serious adverse reaction (SAR). The secondary outcomes will be efficacy as measured by patient-reported outcome measures. Ultrasound techniques will assess the radiological outcome of the procedure. Computerised 'ultrasound tissue characterisation' (UTC) is also carried out in order to add in a less subjective assessment. UTC involves a precision instrument that moves the ultrasound probe automatically across the long axis of the Achilles tendon collecting transverse images at even distances of 0.2 mm over a length of 12 cm and enable tomographic visualisation and tissue characterisation and quantification of architecture and integrity of the collagenous matrix.

Ultrasound examinations undertaken in this study will performed by a musculoskeletal radiologist with experience of undertaking such an examinations. The end

of the Autologous Stem Cells in Achilles Tendinopathy (ASCAT) trial is the last visit of the last participant.

### Inclusion criteria

Patients aged 18–70 years (both males and females) with chronic mid-portion AT (defined by pain and tender swelling in mid-portion of AT, >1 cm proximal to the insertion), with symptoms for longer than 6 months who have failed conservative treatment (at least a full course of physiotherapy) and for whom surgery is being considered. All participants must be able to provide written informed consent. Females of childbearing age and potential must be willing to use two forms of effective contraception from the time of consent to 6 months postinjection (box 1).

### Recruitment and consent

Patients who are referred to the RNOH will be identified as potentially eligible for the trial by the Chief Investigator

## Box 1 Inclusion and exclusion criteria for the study

**Inclusion criteria:**
1. Aged ≥18 and ≤70 years (both males and females).
2. Participants with chronic mid-portion Achilles tendinopathy (AT) (as defined by pain in region of AT and tender swelling in mid-portion of the AT with symptoms for longer than 6 months who have failed conservative treatment (at least a full course of physiotherapy) and for whom surgery is being considered.
3. Able to provide written informed consent.
4. Females of childbearing age and potential must be willing to use two forms of effective contraception from the time of consent to 6 months postinjection.

**Exclusion criteria:**
1. Previous bony surgery (eg, reconstructive pelvic osteotomy) at or in proximity to the bone marrow harvest site.
2. Pregnancy or lactation (self-declaration at screening visit with follow-up urine pregnancy test to be confirmed not beyond 7 days prior to mesenchymal stem cell (MSC) harvest and administration).
3. Current use of steroids, antitumour necrosis factor drugs, methotrexate or ciprofloxacin (or use within 4 weeks of assessment for eligibility).
4. Positive for hepatitis B virus, hepatitis C virus, HIV-1 and HIV-2, syphilis and human T-cell lymphotropic virus.
5. Previous AT surgery on the tendon to receive MSC implantation.
6. Inflammatory arthritis.
7. Known or suspected underlying haematological malignancy.
8. Other active malignancy in the past 3 years.
9. Bovine or antibiotic allergy (since these are in the culture medium).

(CI) or delegated clinician and invited to participate. The RNOH is a tertiary referral centre for AT and potential participants will be referred by letter to the Foot and Ankle Department at the RNOH. The CI or delegated clinicians at the Foot and Ankle Department will review referral letters. In the main, it may not be possible to identify relevant patients with chronic mid-portion AT from the referral letters; however, Patient Information Sheets may be sent by post or email to referred patients informing them of the trial. The CI or delegate will wait until the patient is seen in the initial outpatient consultation to explain the trial in more detail and if they are eligible invite them to participate. Participant Identification Centres (PIC) will also be used to identify potential patients for the trial and these may include primary care musculoskeletal triage services. Potential participants will be referred by the PIC to the trial site.

### Screening assessments
Prescreening interview will include a medical history with discussion about willingness to participate in the trial. This will be followed by a clinical assessment that will include a routine examination of the foot and ankle, including: inspection of limb alignment and other biomechanical abnormality such as flat or high arched feet; size and location of swelling over the Achilles tendon and range of motion of the ankles, hindfeet, midfeet and forefeet. Assessment of gastrocnemius tightness will be performed with the knee fully straight and flexed 20

degrees (Silverskiold test). Examination will aim to rule out other conditions such as an adjacent bursitis or insertional tendinopathy (tenderness over the calcaneal bony attachment). The Simmonds' test will be performed to ensure continuity of the tendon.

Blood tests will be performed to ensure compliance with the Human Tissue (Quality and Safety for Human Application) Regulations 2007 and local procedures. The minimum blood tests required as specified by the regulations will be hepatitis B virus, hepatitis C virus, HIV-1 and HIV-2, syphilis and human T-cell lymphotropic virus. These tests will be carried out at least 1 week before (and within 30 days of the bone marrow harvest procedure) in line with the regulations. Additionally, a pregnancy test will be carried out prior to the bone marrow harvest (within 1 week), if the participant is a female of childbearing age and potential. If any of these tests are positive, the participant will be counselled and deemed ineligible for the trial.

### Baseline assessment
Baseline assessments to be carried out on day 0, immediately prior to harvest include:
▶ Clinical assessment;
▶ Baseline questionnaire on demographics and comorbidity/concomitant medication;
▶ Victorian Institute of Sports Assessment Achilles (VISA-A) Questionnaire;
▶ Visual Analogue Score (VAS);
▶ Manchester Oxford Foot and Ankle Questionnaire (MOXFQ);
▶ EuroQol 5-dimension 5-level (EQ-5D-5L) Questionnaire;
▶ Sporting Activity Score (SAS);
▶ Ultrasound imaging and UTC;
▶ Adverts events (AE) review.

Table 1 shows all the planned assessments and procedures that will be performed before MSC implantation.

### Treatment procedure
For eligible participants, approximately 8 mL of bone marrow aspirate will be collected from the posterior iliac crest into a heparinised syringe under either local or general anaesthetic. A decision on which technique will be used will depend on discussions with the patient. MSC processing and manufacturing will be at the cell-processing centre at the Centre for Cell Gene and Tissue Therapeutics based at the Royal Free Hospital in London. The cells are cultured using a standardised protocol using fetal calf serum in the culture medium with robust validation systems to confirm quality and sterility of cells. Approximately 5 weeks after bone marrow harvest, participants will undergo implantation of the culture expanded MSCs under local anaesthetic in the outpatient setting following clinical assessment and tendon imaging using conventional ultrasound and UTC measurement.[27]

A single injection of between 4 and 20×10[6] MSCs suspended in 1 mL of Dulbecco's Modified Eagle Medium

**Table 1** Schedule for assessments

| | Screening | Bone marrow harvest* | Baseline (immediately before implantation) | Implantation | Follow-up | | | |
|---|---|---|---|---|---|---|---|---|
| Visit number | 1 | 2 | 3 | | 4 | 5 | 6 | 7 |
| Timing postimplantation | | Approximate week: 5 | Day 0 | | Day 2† | Week 6 | Week 12 | Week 24 |
| Deviation window (days) | ‡ | ±14 | 0 | | 0 hours | ±7 | ±7 | ±7 |
| Eligibility review | X | | X | | | | | |
| Medical history | X | | | | | | | |
| Informed consent | X | | | | | | | |
| Clinical assessment | X | | X | | | | | |
| HBV, HCV, HIV-1 and HIV-2 and syphilis tests | X | | | | | | | |
| Pregnancy tests | X§ | X | X¶ | | | | | |
| Bone marrow harvest | | X | | | | | | |
| MSC implantation | | | | X | | | | |
| Routine course of physiotherapy | | | | | | X** | X** | X** |
| Ultrasound imaging (conventional ultrasound and UTC) | | | X** | | | X†† | X†† | X†† |
| VISA-A Questionnaire | | | X | | | X | X | X |
| VAS | | | X | | | X | X | X |
| MOXFQ | | | X | | | X | X | X |
| EQ-5D Questionnaire | | | X | | | X | X | X |
| SAS Questionnaire | | X | | | | | | X |
| Concomitant medication | X | X | X | X | X | X | X | X |
| AE review | | X | X | X | X | X | X | X |

*Urine pregnancy test at bone marrow harvest.
†Day 2 follow-up assessments are done by telephone.
‡Screening procedures must take place at least 1 week before and within 30 days of the bone marrow procedure.
§Self-declaration from the patient.
¶Within 7 days of implantation.
**The number of visits are dependent on participant's progress and are not obligatory at these time points. Standard NHS protocol for physiotherapy will start within 1–2 weeks after implantation. Dates of physiotherapy attendances will be recorded in the case report form.
††Two radiologists will independently scan the participant using ultrasound. UTC will also be measured twice on each occasion.
AE, adverse events; EQ-5D-5L, EuroQol 5-dimension 5-level; HBV, hepatitis B virus; HCV, hepatitis C virus; MOXFQ, Manchester Oxford Foot and Ankle Questionnaire; SAS, Sporting Activity Score; UTC, ultrasound tissue characteristics; VAS, Visual Analogue Score; VISA-A, Victorian Institute of Sports Assessment Achilles Questionnaire.

will be injected under ultrasound control by a delegated, experienced and trained radiologist along the length of the tendon at the area of greatest degeneration starting just distal to the degeneration and finishing just proximal to normal structure. Participants will be observed for a minimum of 2 hours after implantation for any immediate adverse effects.

All participants will be referred for a standard course of physiotherapy. The physiotherapist will issue a discharge summary following their treatment and a copy of this will be included in the participant record. This summary includes details on participant compliance.

### Subsequent assessments

Subsequent assessments following the MSC implantation include AE review, VAS (pain), MOXFQ, VISA-A Questionnaire, EQ-5D, SAS Questionnaire, ultrasound imaging (conventional ultrasound and UTC) and clinical assessment (table 1). These will occur at 6, 12 and 24 weeks post-MSC implantation. After 2 days, the trial

coordinator will phone the patient to check they are well and record any AEs on the ASCAT case report form. The trial will end on the last visit of the last precipitant.

## Patient and public involvement

Patients and the public were not directly involved with the development of the current protocol. The primary research question is the safety of use of culture expanded MSCs. This will enable the development of a larger RCT that will involve oversight from the patient and public representatives. Patient recruitment is described above and the results will be disseminated to the participants via a news letter.

## Power and sample size

Sample size was determined to ensure a predetermined level of accuracy for the incidence of SAR using an exact binomial CI. With 10 participants, if no SAR is observed we can rule out a true SAR rate of 31% or greater with 95% confidence, or a true SAR rate of 21% or greater with 80% confidence, or a true SAR rate of 13% or greater with 50% confidence. With six participants for the interim analysis, if no SAR is observed we can rule out a true SAR rate of 46% or greater with 95% confidence, or a true SAR rate of 32% or greater with 80% confidence, or a true SAR rate of 21% or greater with 50% confidence. Since the study is a proof-of-concept pilot study, no formal power calculation was performed.

## Statistical analysis

The findings from this study will be published and the data will also be included in subsequent larger trials if appropriate. Baseline data collected and summarised will include: participant gender, age, height, weight, body mass index, duration of symptoms, sporting activity, smoking/non-smoking, medication, any other musculoskeletal problems, EQ-5D, MOXFQ, VAS (pain), SAS score, VISA-A score and ultrasound findings (conventional ultrasound and UTC findings).

Continuous baseline data will be presented as mean and SD for normally distributed variables, medians and IQRs for non-normally distributed variables and frequencies and percentages for categorical baseline variables. For the primary outcome, the proportion of participants experiencing a SAR at any time over the 24-week follow-up period will be calculated along with exact binomial CIs to express the uncertainty in the observed incidence rate of SAR.

Since the primary outcome is the incidence of SARs, missing outcomes are not anticipated. The primary analysis will be as per-protocol. All participants will receive the same treatment, which cannot be withdrawn once given and we will only analyse the data from the participants who actually receive autologous culture expanded MSCs. Cancellation of implantation will be a logistical rather than surgical decision and as such is not expected to introduce bias. The proportion of participants experiencing a success at 6 months will be calculated along with exact binomial CIs to express the uncertainty in the observed incidence of success at 6 months. Changes from baseline in EQ-5D, MOXFQ, VAS (pain), VISA-A and SAS scores will be summarised as mean and SD if normally distributed, or median and IQR if the data do not demonstrate normality.

## ETHICS AND DISSEMINATION

The findings of this study will be disseminated through peer-reviewed publications and conference presentations.

**Contributors** AJG, RZ, RS, LM, DB, MK, RG, LK and POD developed the trial protocol and follow-up schedule. LK and MK performed the literature review prior to grant submission. AJG, RZ and DB all contributed to the writing of the manuscript and finalising prior to submission. AJG is the chief investigator for the study.

**Funding** The trial is funded by the UK Stem Cell Foundation (Registered Charity No: 1110009).

**Competing interests** None declared.

**Patient consent** Not required.

**Ethics approval** The protocol has been approved by the National Research Ethics Service Committee (London, Harrow; reference 13/LO/1670).

**Provenance and peer review** Not commissioned; externally peer reviewed.

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
