## [Reviewer comments · BMJ Open]

ARTICLE DETAILS

TITLE (PROVISIONAL)	Autologous Stem Cells in Achilles Tendinopathy (ASCAT): protocol for a phase IIA, single centre, proof of concept study
AUTHORS	Goldberg, Andrew; Zaidi, Razi; Brooking, Deirdre; Kim, Louise; Korda, Michelle; Masci, Lorenzo; Green, Ruth; O'Donnell, Paul; Smith, Roger

VERSION 1 – REVIEW

REVIEWER	Denitsa Docheva Experimental Trauma Surgery Department of Trauma Surgery University Regensburg Medical Centre Regensburg, GERMANY
REVIEW RETURNED	26-Jan-2018

GENERAL COMMENTS	The study protocol is well design and written. Two questions: the autologous MSC are they expanded in culture medium containing autologous serum? Studies have suggested that such has benefits in terms of MSC expansion in comparison to bovine serum. This may reduce the expansion and waiting for treatment period of 5 weeks. Are the cells validated for quality and lack of contamination (for example mycoplasma) after expansion?
---

REVIEWER	Federico Uselli IRCCS Istituto Ortopedico Galeazzi, Milan, Italy
REVIEW RETURNED	01-Mar-2018

GENERAL COMMENTS	The protocol is interesting but major revision are requested before publication Introduction: Page 5: L11: add reference L23: add reference L39: add reference Page 6: L7: please reformulate the sentence using the following citation. Intratendinous adipose-derived stromal vascular fraction (SVF) injection provides a safe, efficacious treatment for Achilles tendinopathy: results of a randomized controlled clinical trial at a 6-month follow-up. Uselli FG, Grassi M, Maccario C, Vigano' M, Lanfranchi L, Alfieri Montrasio U, de Girolamo L. Knee Surg Sports Traumatol Arthrosc. 2017 Mar 1. doi:
--

	10.1007/s00167-017-4479-9. L51: same as above Methods: page 8: L26: i think this is the weakness of the study. 10 partecipanti without control group. L34: the only use of US is another weakness of the study. Is very operator dependent. L43:patients aged 18 or 70 are very different . i think age should be reduced Screening assessments: the use of MRI should be suggest
--	---

VERSION 1 – AUTHOR RESPONSE

Reviewer 1

Reviewer Name: Denitsa Docheva

Institution and Country: Experimental Trauma Surgery, Department of Trauma Surgery, University Regensburg Medical Centre, Regensburg, GERMANY

Please state any competing interests or state 'None declared': Nothing to declare.

Please leave your comments for the authors below

The study protocol is well design and written. Two questions:

1. the autologous MSC are they expanded in culture medium containing autologous serum? Studies have suggested that such has benefits in terms of MSC expansion in comparison to bovine serum. This may reduce the expansion and waiting for treatment period of 5 weeks.

The cells are cultured in fetal calf serum and do not use autologous serum. Much of the work pioneering autologous serum culture was carried out by Roger Smith (an author on this paper) who pioneered the stems cell technique in horses.

In fact for their horse treatment (which has been used on thousands of horses) in the main uses fetal calf serum. When they considered using the technique for joints they still cultured the cells in fetal calf serum but for the last 24 hours the serum is changed to medium containing autologous serum. The purpose of this was to reduce the antigenicity of the cells.

There is no strong evidence of the benefits of autologous serum over fetal calf serum in terms of efficacy. When this study was discussed with the MHRA (regulator) it was suggested that we used the standardised treatment and in addition our cell processing laboratory which is highly regulated would not be able to have changed their existing techniques without going through costly and time consuming renewed validation steps.

A discussion on the merits and risks of using human autologous serum is considered beyond the scope of this protocol paper, but we have now amended the paper to make it clear that the cells are not cultured in autologous serum.

2. Are the cells validated for quality and lack of contamination (for example mycoplasma) after expansion?

Yes the laboratory is highly regulated and the paper has now been amended to reflect this.

Reviewer 2

Reviewer Name: Federico Uselli

Institution and Country: IRCCS Istituto Ortopedico Galeazzi, Milan, Italy

Please state any competing interests or state 'None declared': personal fees from Integra and Geistlich, and grants and personal fees from Zimmer, outside the submitted work.

Please leave your comments for the authors below

The protocol is interesting but major revision are requested before publication

Introduction:

Page 5:

L11: add reference

More than 85,000 patients present to primary care with Achilles Tendinopathy (AT) annually (General

Practitioner Research Database 2007).

This UK database was manually checked by our team and only refers to those coded in primary care. This database is not referenceable with a peer reviewed journal. As such we have amended the paper with a referenceable source as to incidence, as follows:

The general population has an incidence of 2.35 per 1000 people, equivalent to more than 150 000 people in the United Kingdom every year (reference de Jonge S, van den Berg C, de Vos RJ, et al. Incidence of midportion Achilles tendinopathy in the general population. Br J Sports Med 2011;45:1026–1028.)

L23: add reference

This line has now been referenced

L39: add reference

This line has now been referenced

Page 6:

L7: please reformulate the sentence using the following citation.

Intratendinous adipose-derived stromal vascular fraction (SVF) injection provides a safe, efficacious treatment for Achilles tendinopathy: results of a randomized controlled clinical trial at a 6-month follow-up. Usuelli FG, Grassi M, Maccario C, Vigano' M, Lanfranchi L, Alfieri Montrasio U, de Girolamo L. Knee Surg Sports Traumatol Arthrosc. 2017 Mar 1. doi: 10.1007/s00167-017-4479-9.

We thank the reviewer for this reference which we have now referenced in both areas highlighted.

L51: same as above

Methods:

page 8:

L26: i think this is the weakness of the study. 10 partecipanti without control group.

This study is designed as a First In Man safety study. In a First in Man study the primary outcome is safety. Efficacy is assessed but the study is not powered to assess efficacy and no control group is used. The other purpose of this study is to provide important data to enable appropriate powering of a subsequent Phase III study in which efficacy is assessed against a control group. This includes the

range of scores and standard deviation of all of the clinical scores captured. The reviewer is entirely correct in stating that larger numbers will be needed and comparison with a control is needed, but this study is an important stepping stone to that end.

We have made this point much clearer now in the text.

L34: the only use of US is another weakness of the study. Is very operator dependent.

We entirely agree with the reviewer that ultrasound is very operator dependant and this is why we have used UTC in addition to conventional ultrasound. (van Schie HT, de Vos RJ, de Jonge S, Bakker EM, Heijboer MP, Verhaar JA, Tol JL and Weinans H. Ultrasonographic tissue characterisation of human Achilles tendons: quantification of tendon structure through a novel non-invasive approach. Br J Sports Med 2010; 44(16): 1153-1159. PMID: 19666626 Download: UTC-vanSchie 2010)

L43:patients aged 18 or 70 are very different . i think age should be reduced

We entirely agree with the reviewer. At the time of the ethics application we stated 18-70 on the basis that we wanted to assess the ability of stem cells to grow in different aged patients. In fact we have carried out preliminary culture now of many patients of differing age groups and can concur with the reviewer that the cells of patients over 50 do not replicate well and this leads to failure to expand cells in a reasonable time frame. As such we think that the age for inclusion in any future studies should be patients aged under 50.

However, given that we have ethical and regulatory approval for the protocol to include ages up to 70 we have left this in the protocol paper. It will be our intention when we write up the results of the First In Man safety study to discuss this point and provide documentary evidence that the reviewer is entirely correct in his assumption.

Screening assessments:

the use of MRI should be suggest

Many of the patients have MRI scans routinely at our centre as part of standard care and so we will have that information as well. For the purposes of the First In Man study we elected not to include MRI assessment as we wanted to compare conventional ultrasound (carried out by two consultant radiologists) with UTC and in order to keep costs and logistics manageable we elected not to analyse MRI scans as well.

VERSION 2 – REVIEW

REVIEWER	Denitsa Docheva Experimental Trauma Surgery, Department of Trauma Surgery, University Regensburg Medical Centre, Regensburg, GERMANY
REVIEW RETURNED	28-Mar-2018

GENERAL COMMENTS	The authors addressed the reviewer's questions.
REVIEWER	federico usuelli istituto ortopedico galeazzi italy
REVIEW RETURNED	29-Mar-2018
GENERAL COMMENTS	authors have answered all the requested reviews, so i suggest to accept the protocol